# Development and Modeling of an Onion Harvester with an Automated Separation System

**Michel N. Erokhin, Alexey S. Dorokhov, Alexey V. Sibirev \*, Alexandr G. Aksenov, Maxim A. Mosyakov, Nikolay V. Sazonov**  **and Maria M. Godyaeva**

Federal Scientific Agroengineering Center VIM, Moscow 109428, Russia; n.erohin@rgau-msha.ru (M.N.E.); dorokhov.vim@yandex.ru (A.S.D.); 1053vim@mail.ru (A.G.A.); maks.mosyakov@yandex.ru (M.A.M.); sazonov_nikolay@mail.ru (N.V.S.); airrune@yandex.ru (M.M.G.)

\* Correspondence: sibirev2011@yandex.ru

**Abstract:** One of the most important problems during the implementation of any technology is to reduce labor costs, energy, and resource conservation while increasing the yield of cultivated crops and, as a result, reducing the cost of production. Despite a significant amount of scientific research devoted to the problem of energy and resource conservation in the cultivation and harvesting of agricultural crops and the development of mechanization tools that ensure the high-quality performance of technological operations, there remain issues that have not been fully resolved to date. In addition, not all the results of known theoretical and experimental studies can be directly applied to intensify the process of harvesting root crops since the quality indicators of marketable products depend on the type and technological parameters of the separating working bodies. This article presents the design of a rod elevator with an adjustable angle of inclination of the web, which reduces damage to commercial products of root crops and bulbs with maximum completeness of separation. A laboratory facility has been developed to substantiate the design and technological parameters of a separating system with an adjustable web inclination angle. Based on the results of theoretical and experimental studies, a machine for harvesting onions with an adjustable blade inclination angle has been developed, which provides an increase in the quality indicators of onion harvesting at optimal values of the parameters: (1) translational speed of movement of the rod elevator with an adjustable web inclination angle of 1.7 m/s with a 98.4% completeness of separation and 1.7% damage to the bulbs; (2) translational speed of the movement of the machine for harvesting root crops and onions 1.0 m/s with a 98.5% separation completeness and 1.1% damage to the bulbs; (3) digging depth of the digging plowshare equal to 0.02 m, with an onion heap separation completeness of more than 98% and product damage of less than 1.4%. The results of theoretical and experimental studies of a rod elevator to substantiate the design and technological parameters during its interaction with a heap of onion are presented. Basic design and technological parameters of the studied rod elevator are substantiated, namely, the distance S1 of the movement of the rod of the actuators, the angle a1 of the longitudinal inclination of the surface of the rod elevator relative to the horizon, and differential equations of motion of the onion-sowing pile element on the surface of the rod elevator with an adjustable angle of inclination of the web.

**Keywords:** technological process; rod elevator; lifting angle; displacement; experiment; cleaning machine



## 1. Introduction and Literature Review

Modern agricultural enterprises are the largest consumers of all resources, including labor and energy. The most energy-intensive branch of agriculture remains crop production, which accounts for 70% of all costs, including more than 40% for operations related to harvesting.

The most important problem in the implementation of any technology is to reduce labor costs, energy, and resource conservation while increasing the yield of cultivated crops and, as a result, reducing the cost of production.

Onions are a valuable food crop. In recent years, in Russian agricultural production, onion has occupied a leading place in terms of the sown area, along with white cabbage. Every year, they are sown within areas ranging from 88 to 96 thousand hectares. There is a constant increase in large-scale production; the share of professional production fluctuates over the years and reaches 40–50%. The most common and most mastered method used in Central Russia, as well as in the northern part of European countries, is the cultivation of turnips from sets. At present, harvesting remains the most resource-intensive technological process of onion production.

This circumstance is due to the increased rise in the soil layer during the extraction of bulbs from the soil and further cleaning of the heap from soil and plant impurities on the separating working bodies. Since the bulbs are small in size, the process of separation from soil impurities on the separating working bodies proceeds unsatisfactorily and the final cleaning from soil impurities is carried out during post-harvest processing with extensive use of manual labor. Despite the existence of extensive research on the issue of mechanized onion harvesting technology, there are unresolved problems in this area, which, in most cases, are associated with the imperfection of the design of the digging and separating organs of machines for harvesting onion sets, as evidenced by the content in the heap to be divided commensurate with bulbs of onion sets in clods of soil.

Various kinds of intensifiers for the separation of onion sets significantly injure them, due to the forces from interaction with the separating working bodies and directly with the bulbs. To increase the efficiency of onion harvesting, it is necessary to destroy soil clods before they enter the receiving and digging part of the onion harvesting machine, which ensures a reduction in damage (injury) of the bulbs, a reduction in energy costs while increasing the completeness of separation of the onion heap with gentle mechanical chemical effect on the bulbs and, as a result, an increase in the productivity of the harvesting process.

Therefore, the creation of technical means to improve the efficiency of onion harvesting machines is an urgent task in the seed production and commodity production of onions, the solution of which will reduce the cost of seed material during post-harvest processing.

The analysis of the designs of existing machines for harvesting root crops and onions, as well as patent and technical literature, presented below, revealed shortcomings in the design of the separating working bodies of machines for harvesting root crops and bulbs [1], which do not allow for high-quality performance of the technological process of the separation of marketable products.

A potato harvesting machine [2] consists of three main elements, namely: 1, a digging ploughshare, 2, a separating rod elevator, and 3, supporting rods (Figure 1). The main distinguishing feature of the harvesting machine presented above is 1, the execution of the digging working body (Figure 2) and 3, the separating rod elevator (Figure 3).

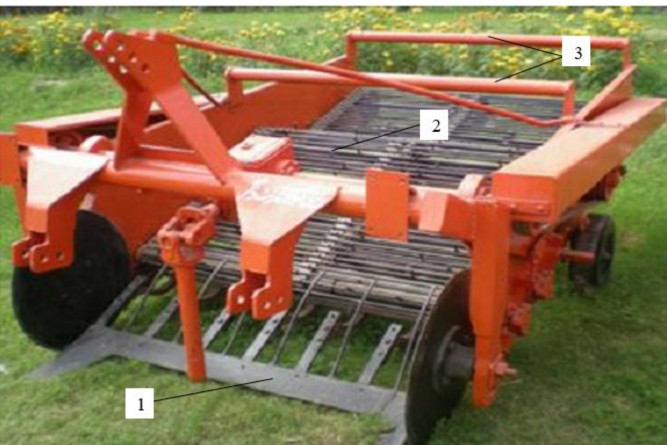

**Figure 1.** General view of a machine for harvesting root crops: 1, scrounging; 2, separating rod elevator; 3, supporting rods.

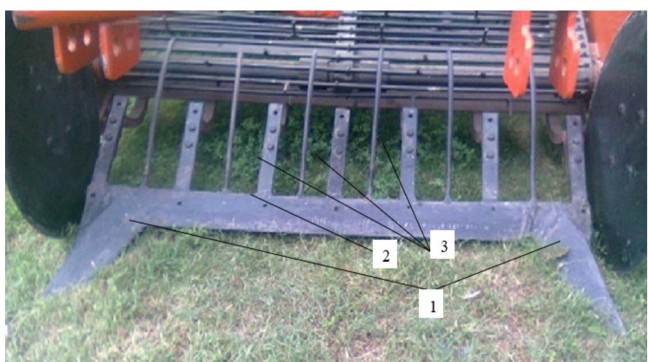

**Figure 2.** General view of the sub digging body: 1, ripping element; 2, digging element; 3, slotted pre-separation apertures.

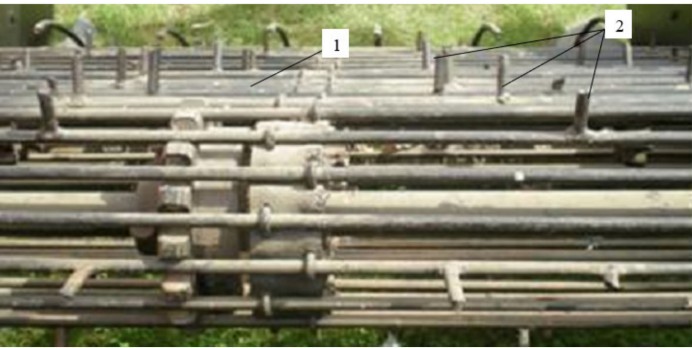

**Figure 3.** General view of the separating rod elevator: 1, rod; 2, kernel.

The digging working body is a combination of loosening (1) and digging (2) elements with slotted holes (3) of preliminary separation formed by the working surface of the digging element (2).

The loosening element (1) of the digging working body performs preliminary loosening of the excavated soil layer in order to reduce the flow of soil lumps to the separating working bodies, and, therefore, intensifying the process of cleaning root crops from commensurate soil lumps, as well as reducing the amount of damage to commercial products from collisions with soil lumps on the rod elevator of the harvesting machine. The digging element (2) extracts root crops from the soil with pre-separation through the slotted holes (3) of pre-separation.

The disadvantages of this digging body include increased losses of root crops through the slotted holes (3) of the digging element (2); in addition, this structural design of the digging working body does not reduce the amount of damage to the separated products on the rod elevator due to insufficient loosening of the soil layer at the location of the root crops.

To intensify the separation process of a pile of root crops, the separating surface of the rod elevator is made with rods (2) located on bars (1) (Figure 3).

The main purpose of the rods (2) located on the rods (1) is the destruction of soil lumps coming from the digging working body of the harvesting machine. However, in addition to the process of destruction of the soil lumps directly, there is an intense force effect of rods (2) on root crops, which increases damage to the separated products.

Also, the design of the separating rod elevator (Figure 4), which is a separation intensifier, consists within it a passive two-shoulder shaker (4) located under the upper branch of the web of the rod elevator (3) [3].

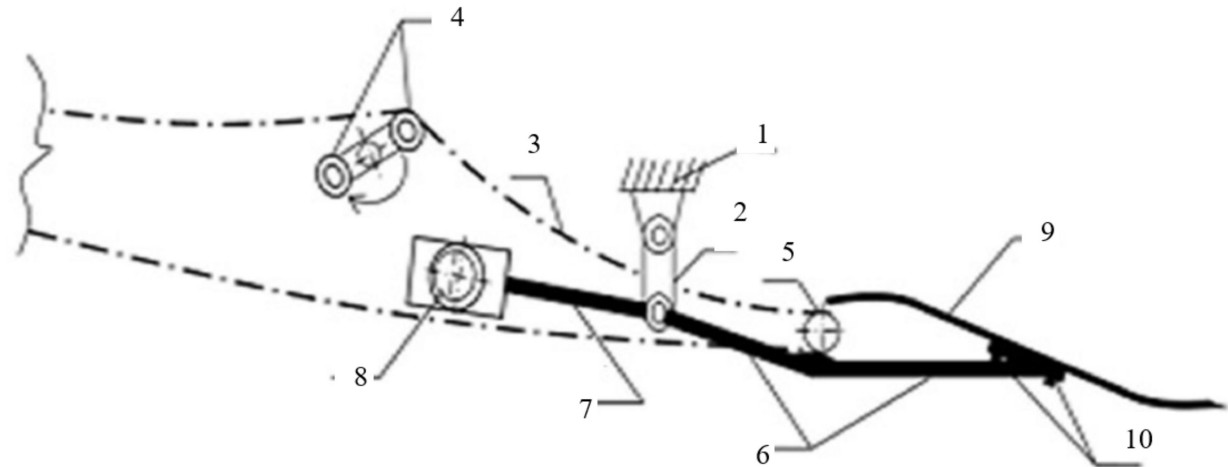

**Figure 4.** Scheme of separating rod elevators: 1, frame; 2, thrust; 3, rod elevator; 4, double-shoulder shaker; 5, support roller; 6, mounting bracket; 7, intermediate bracket; 8, eccentric shaft; 9, scrounging; 10, bracing.

In addition, the front part of the web of the rod elevator from the side of the digging plowshare (9) oscillates in the vertical plane when the mounting bracket (6) of the digging plowshare (9) is exposed to the supporting roller (5), which provides an additional force on the soil layer, thereby intensifying the soil and plant impurities separation process.

The disadvantages of the well-known design of the rod elevator include increased damage to root crops during the transition from one cascade to another, as well as the inability to disperse the pile of root crops over the entire width of the conveyor [4].

Analysis of the technical means of mechanized root crop harvesting allows us to conclude that the functioning elements of the harvesting machine with various types of separation intensifiers do not provide high-quality indicators of root crop harvesting for such indicators as the completeness of separation and the damage to root crops.

It is known that at the stage of the second phase of harvesting root crops and onions, machines whose technological process repeats the stage of the first phase of harvesting are used, and then a contradictory situation arises due to the constancy of the regime and technological parameters of the harvesting machine at all stages of onion harvesting, which are unacceptable for the work of the second phase of harvesting, thus leading to bulb damage [5].

This circumstance is because, on the one hand, a decrease in the depth of the ploughshare in the soil at the stage of selecting bulbs from rolls leads to a decrease in the rise together with bulbs of soil lumps commensurating with them, which contributes to improving the quality of onion separation.

On the other hand, due to a decrease in the amount of soil entering the digging and separating organs, damage to onion bulbs occurs as a result of a decrease in the soil layer between the working surface of the elevator and the products being cleaned; in addition, rod elevators do not always provide high-quality separation of soil impurities [6–8].

The need to reduce the traction and specific traction resistances of agricultural implementations require a constant search for more advanced working bodies and fieldwork technologies in terms of energy intensity. The use of working bodies of one type or another causes the predominance of a certain type of deformation of the soil layer [9–11]. Since the soil does not equally resist different types of influences, this leads to a decrease or increase in its resistivity. At the same time, the whole power characteristic is based on the force $R_\Pi$-the traction resistance of the organ [12–14].

In order to improve the quality of separation of a pile of root crops and onions, various types of passive or active rod elevator shakers are used in the construction of rod elevators. The kinematic model of the oscillation of the working branch of the rod elevator is assumed

such that the bulbs are thrown on the canvas of the elevator and fall on the bars, causing them to collide with the bars and result in damage due to the vertical component of the gravity of the bulb [15–17].

To reduce the destructive effects of separating working bodies of machines on bulbs, it is necessary to design a separating device that intensifies the cleaning process and minimizes the impact of the vertical component of the gravity of the bulb [18–20].

It is known that the separating capacity of a rod elevator depends on the angle of inclination $\alpha$ and the speed $v_{EL}$ of the elevator movement (Figure 5).

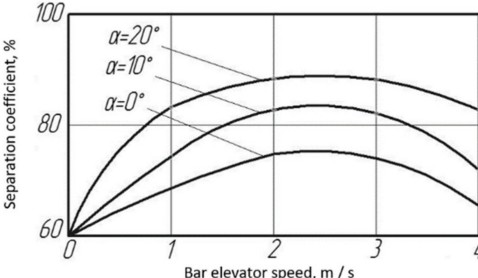

**Figure 5.** Graph of the separation coefficient $\varepsilon$ of the rod elevator from the angle $\alpha$ of the inclination of the rod elevator and translational speed $v_{EL}$ [4].

The separation coefficient $\varepsilon$ at any angle of inclination $\alpha$ rises with an increase in the speed $v_{EL}$ of the movement of the rod elevator to a certain maximum value, after which it begins to decrease. The optimal speed $v_{EL}$ of the rod elevator movement is an interval of 2–2.5 m/s.

In addition, in order to exclude twisting of a pile of root crops and onions on the surface of a rod elevator, it is necessary to comply with the conditions [21,22]:

$$v_{EL} = v_{Me} = v_K \cdot A \tag{1}$$

$v_K$—translational speed of the harvesting machine, m/s; A—is the stable coefficient (A = 1.3).

The text of the article presents the following designations, displayed in Table 1.

**Table 1.** Name of symbols.

| № п/п | Designation | Name |
|---|---|---|
| 1 | $v_{EL}$ | forward speed of the rod elevator, m/s |
| 2 | $v_K$ | forward speed of the harvester, m/s |
| 3 | m | mass of the onion-soil pile, kg |
| 4 | $v_L$ | translational speed of the harvesting machine, m/s |
| 5 | $L_L$ | length of the digging plowshare, m |
| 6 | $L_{EL}$ | length of the rod elevator, m |
| 7 | $D_{VED}$ | diameter of the driven shaft of the rod elevator drum, m |
| 8 | $D_{PR}$ | diameter of the drive shaft of the rod elevator drum, m |
| 9 | $T_N$ | operating time under load, s |
| 10 | $T_O$ | shutdown period, s |
| 11 | $(T_N + T_O)$ | total duration of the working cycle, s |
| 12 | $F_1, F_2, F_3$ | cubic strain, N |
| 13 | $S_1, S_2, S_3$ | rod stroke, m |

The purpose of the study is to develop a harvesting technology, the implementation of which eliminates or significantly reduces the intake of soil lumps together with root crops and bulbs, and in the case of increased intake of a pile of root crops and onions to

the separating working bodies, it is necessary to regulate the regime and technological parameters of the functioning elements of the harvesting machine, depending on the changing soil and climatic conditions of harvesting. The solution to the problem of improving the quality of onion harvesting is seen on the one hand in the extensive development and improvement of technical means that contribute to improving the quality indicators of the technological process of onion harvesting, which leads to an increase in their material consumption due to the mechanical increase in the mass of separating devices [23]. On the other hand, this is in increasing the level of intelligence and responsiveness of the functioning elements of the harvesting machine to changing environmental conditions and regulating their technological and operational parameters.

Scientific novelty is the technology of harvesting root crops and onions with the development of a separating rod elevator with an adjustable angle of inclination of the web.

## 2. Materials and Methods

The separating rod elevator (Figure 6) with asymmetrically installed elliptical shakers (patent No. 2638190) [24] and an adjustable tilt angle (patent No. 2679734) [25] provides a reduction in damage and an improvement in the quality of the separated products as a result of minimizing the impact of the vertical component of root crops and bulbs gravity, as well as increasing the uniformity of the distribution of the pile of root crops and bulbs on the separating surface when the angle of inclination of the rod elevator changes due to the changes in soil and climatic conditions of harvesting root crops and onions [26].

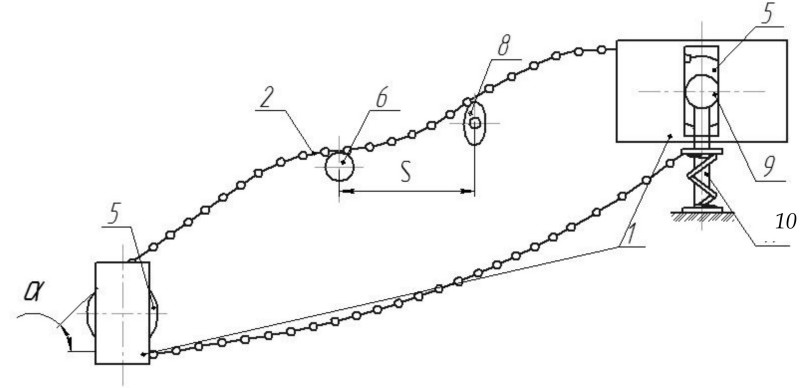

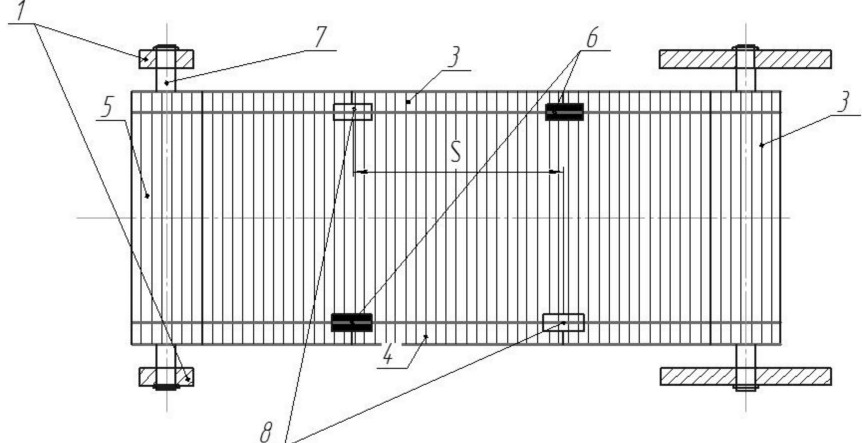

**Figure 6.** Diagram of a separating rod elevator: 1, frame; 2, rod elevator; 3–4, branches of a rod elevator; 5, leading roller; 6, supporting roller; 7, driven roller; 8, elliptical shaker; 9, a shaft of a conducted roller; 10, a mechanism for regulating vertical movement.

Separating conveyor machines for harvesting onion contains mounted on the frame (1) rod Elevator (2), (3), and (4), which is equipped with leading (5), support (6), and the driven rollers (7) mounted on the frame (1).

Under the opposite parties (3) and (4) rod Elevator (2) is set to scramblers (8) offset axes of rotation in the horizontal plane along the length on size S and mismatch of the phases of the lifting and lowering of the opposite parties (3) and (4) rod Elevator (2).

To adjust the angle $\alpha$ of the tilt of the rod elevator within 15–23°, the shaft (9) of the driven roller (7) moves in the groove (10) of the frame (1) on the mechanism (11) for regulating vertical movement.

With this arrangement of the shakers (8) on the separating elevator, the operating mode is provided, in which the tuber-bearing pile moves along the surface of the rod elevator (2) without tossing.

At the moment of lowering the side (3) of the rod elevator (2), the opposite side (4) is lifted along the length S of the rod elevator, i.e., the opposite sides (3) and (4) of the rod elevator (2) work in the opposite phase.

In this case, the probability of damage to the bulbs is less, and the separation quality is better since the time of contact of the bulb with the surface of the rod elevator will be longer.

At the same time, an increase in the quality indicators of onion-sowing separation as a result of the execution of the driven roller (7) on the vertical movement mechanism (11) is provided by changing the angle $\alpha$ of the inclination of the rod elevator (2), depending on the size of the supply of the pile of onion-sowing from the digging working bodies at a constant translational speed of the rod elevator (2) since the separation coefficient increases proportionally to the angle $\alpha$ of the inclination of the rod elevator (2).

In addition, the variation of the angle $\alpha$ of the inclination of the rod elevator, depending on the size of the supply of the onion-sowing pile, provides regulation of the uniformity of the supply of the pile to the secondary separation devices at a constant translational speed of the rod elevator, which leads to increased productivity and completeness of separation during the operation of the onion-sowing machine.

To adjust the angle of inclination of the web of the rod elevator (1), weight sensors (2) are used, mounted on the digging ploughshare (3) (Figure 7).

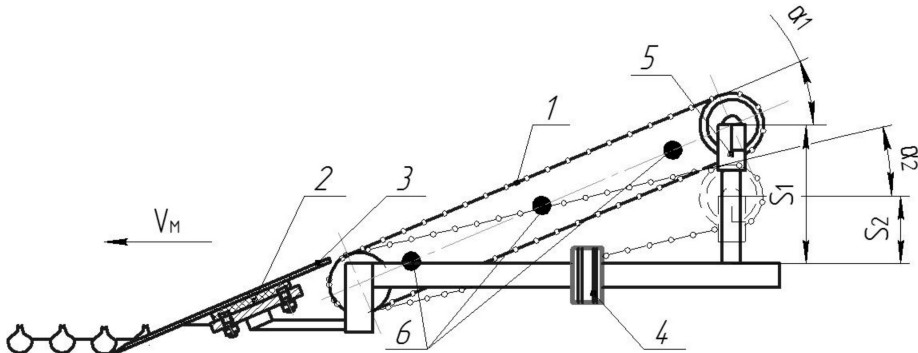

**Figure 7.** Diagram of changes in the angle of inclination of the rod elevator web: 1, rod elevator; 2, weight sensor digging share; 3, digging coulter; 4, microcontroller; 5, electric cylinder; 6, inertial sensor.

Weight sensors (2) in the process of technological operations of separation piles of root vegetables and onions and regulation of the angle of inclination $\alpha 1$ rod Elevator (1) measured by the weight of the incoming crop of roots at undermining the ploughshare (3) and transmit readings to the microcontroller (4).

Linear actuator submitted by electroculogram (5), depending on sensor readings of weight (2), receiving the signal from the microcontroller (4), moves the rod linear actuator to the required distance, or $S_1$ $S_2$, $\alpha_1$ changing the angle of incline rod Elevator (1).

When a pile of root crops passes over the surface of the rod elevator (1), the soil is sifted through the slit holes formed by adjacent rods of the web.

Registration of the sifted soil is carried out by means of inertial sensors (6) installed along the length of the rod elevator (1).

When evaluating the efficiency of separation by the inertial sensor (6), the oscillation frequency will also be characterized by the speed of movement of the rod elevator (1) (change of gaps and bars above the sensor).

However, the amount of sifted soil will be characterized by peak amplitude values. Therefore, the efficiency of elevator separation is evaluated by amplitude values.

If the sifting of the soil on the rod elevator is carried out below the required value set by the microcontroller (4), the angle $\alpha_1$ of the slope of the web of the rod elevator (1) is adjusted in the set range of values.

After determining the mass of the pile of root crops and onions on the digging plowshare, the controller with a time delay T, with the movement of the rod elevator, transmits a control signal to move the rod of the actuators.

$$Q_S = \frac{m \cdot v_L}{L_L} \tag{2}$$

m, mass of the onion-soil pile; kg; $v_L$, translational speed of the harvesting machine; m/s; $L_L$, length of the digging plowshare; m.

Since the adjustment of the angle $\alpha_1$ of the inclination of the web of the rod elevator is carried out by moving the rear shaft of the drive drum of the rod elevator with a diameter of $D_{PR}$, the required distance $S_1$ of the movement of the rod of the actuators in the body of the cycle is carried out according to the formula:

$$S_1 = \left[ L_{EL} - \left( \frac{D_{VED}}{2} + \frac{D_{PR}}{2} \right) \right] \cdot \sin \alpha_1 \tag{3}$$

where $L_{EL}$ is the length of the rod elevator, m; $D_{VED}$ is the diameter of the driven shaft of the rod elevator drum, m; and $D_{PR}$ is the diameter of the drive shaft of the rod elevator drum, m.

It is known that for the automated execution of the technological operation of adjusting the angle of inclination of the rod elevator, the most suitable is the use of electronically controlled electric cylinders (linear actuators) mounted on the frame.

The performance of a linear actuator is influenced by many factors specific to the field of application. The most important factors for the evaluation and selection of linear drives of the developed unit are the pulling and pushing force, static and dynamic load capacity, speed, stroke length, turn-on duration, and service life.

The two main characteristics of the drive that determine its power are the force generated (payload) and the speed of movement of the output link (actuator stem). The speed of movement of the actuator rod depends on the applied load and the type of engine used. In this case, the magnitude of the current depends on the drive power.

Linear actuators are designed for intermittent operations. The duration of activation and the utilization factor determine the maximum period of operation of the drive without stopping.

The utilization factor is defined as the amount of time under load versus the total duration of the switch-on.

If the utilization factor is exceeded, the linear actuator may overheat and fail.

The permissible load for DC actuators at a specific utilization factor is expressed as a percentage of the maximum dynamic load capacity:

$$K_{IP} = \frac{N}{(T_N + T_O)} \cdot 100\% \tag{4}$$

where $T_N$ is the operating time under load, s; $T_O$ is the shutdown period, s; and $(T_N + T_O)$ is the total duration of the working cycle, s.

If the actuator for maintaining a given distance between inductors and plants operates according to the following cycle: 5 s work, 5 s pause, 5 s work, 5 s pause, etc., then the utilization factor for this working cycle will be:

$$K_{IP} = \frac{5 + 5}{(5 + 10) + (5 + 10)} \cdot 100\% = 33.3\% \tag{5}$$

The life of the actuator depends on the load, the stroke length, and how often the safety clutch is triggered.

where p is the pitch of the screw, mm; C is the basic dynamic load capacity; S is the pitch of the screw, mm; and F is the average cubic load, mm.

Often the amount of load on the actuators is not constant. To calculate the equivalent load, it is necessary to determine the average constant load F:

$$F = \sqrt[3]{\frac{F_1^3 \cdot S_1 + F_2^3 \cdot S_2 + F_3^3 \cdot S_3 + F_n^3 \cdot S_n}{S_1 + S_2 + S_3 + S_n}} \tag{6}$$

where $F_1$, $F_2$, $F_3$ is cubic strain, H; $S_1$, $S_2$, $S_3$ is rod stroke, m.

The main component of the adaptation system of the working bodies of the rod elevator of the machine for harvesting root crops and onions is a microcontroller for controlling the movement of actuator-linear actuators (Figure 8).

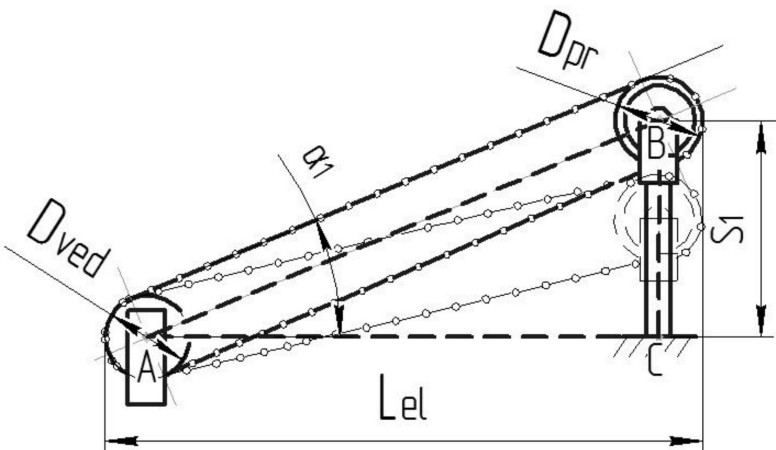

**Figure 8.** Scheme for determining the distance of movement of the actuator rods.

## 3. Results and Discussion

The block diagram of the research methodology is shown in Figure 9.

Based on the theoretical studies carried out, a technology for harvesting root crops and onions with an adjustable angle of inclination of the separating rod elevator was developed, a laboratory installation was developed (Figures 10 and 11), and experimental studies were conducted.

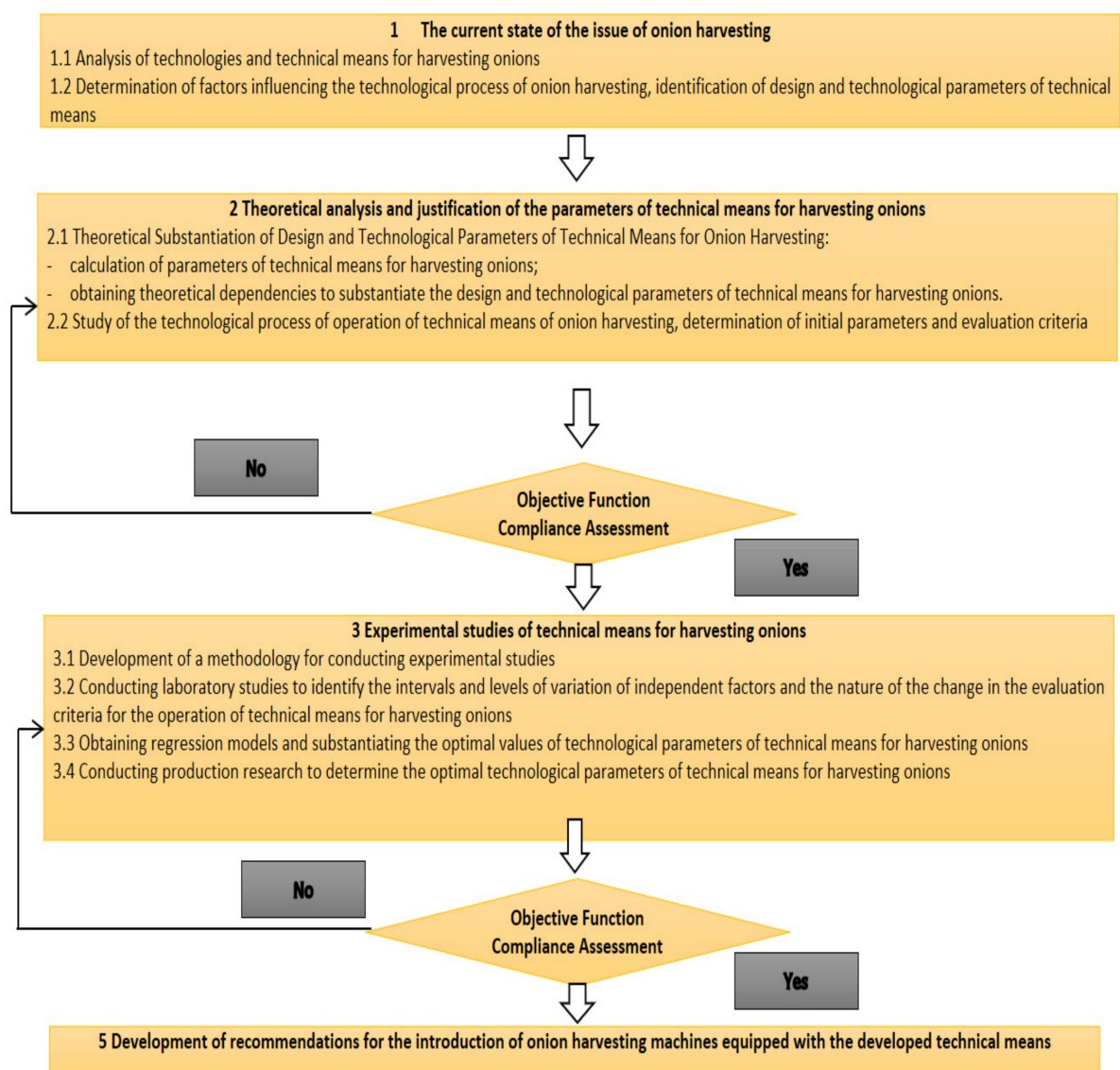

**Figure 9.** Structural scheme of research.

The laboratory installation consisted of a container (2) for the preliminary placement of the pile, a separating rod elevator (3) (working length 1.9 m and width 0.94 m) mounted on the support posts (11) of frame (1).

Passive elliptical shakers (4) and supporting rollers (5) are installed under the web of the rod elevator (2) with the possibility of moving along the frame (1) on the racks (12) by fixing the position with the bracket (13) on the frame (1) with a bolted connection.

The electric actuator rod Elevator (3) is carried out from the electric motor (7), asynchronous brand (N = 0.6 kW; $n$ = 920/1200 rpm) and frequency Converter (9) brand Tecorp Group (N = 0.75 kW; $U_{VX}$ = 220 V), using single-stage reducer 8 (model 1-MC-160-2-23), which are mounted on the base plate (14).

The presence of a frequency converter (9) in the design of the laboratory installation is caused by the need to change the rotation speed of the motor shaft (7). For collecting bulbs after separation, the technological scheme of the laboratory installation provides for the presence of the tarpaulin (6).

The support racks (11) of the rod elevator (3) and the racks (12) of the shakers are made of rectangular steel pipes.

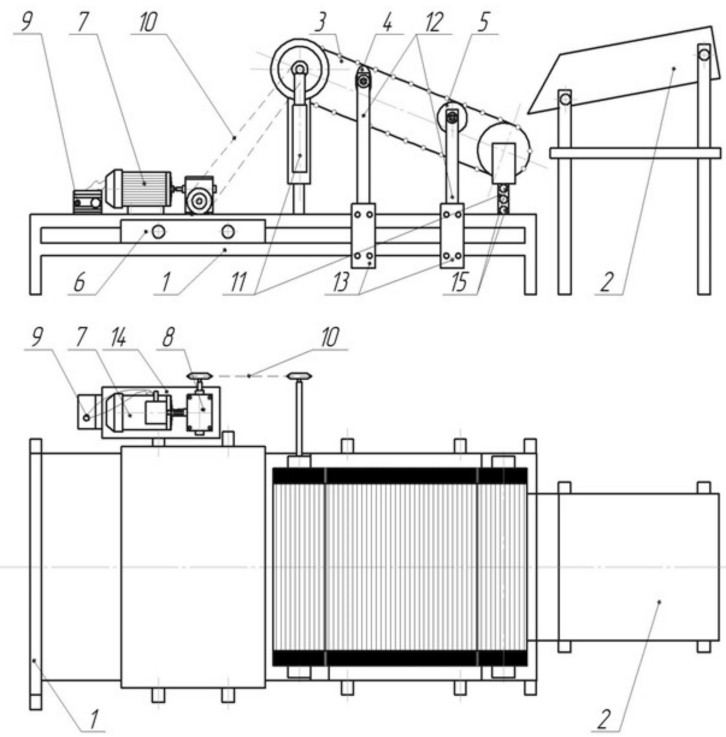

**Figure 10.** Diagram of a laboratory installation for determining the influence of technological parameters of a separating rod elevator on the quality indicators of the separation of a pile of root crops and onions: 1 frame; 2 container for preliminary placement of the pile; 3 separating rod elevator; 4 elliptical shaker; 5 cylindrical roller; 6 tarpaulin of separated products; 7 electric motor; 8 single-stage gearbox; 9 frequency converter; 10 chain transmission; 11 support racks; 12 shaker and support roller racks; 13 connecting bracket; 14 support plate; 15 technological openings.

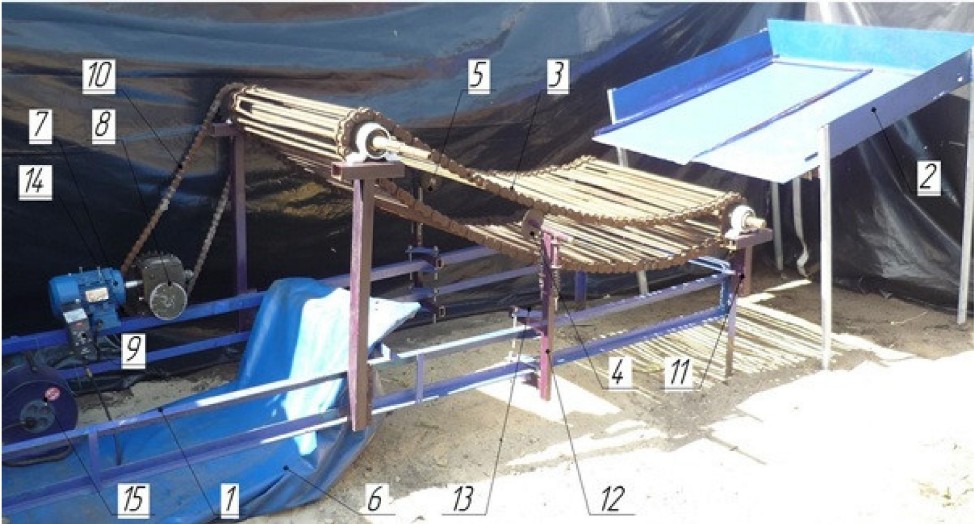

**Figure 11.** A general view of a laboratory installation for determining the effect of the technological parameters of a separating rod elevator on the quality indicators of the separation of a pile of root crops (bulbs): 1 frame; 2 container for preliminary placement of the pile; 3 separating rod elevator; 4 elliptical shaker; 5 supporting roller; 6 tarpaulin of separated products; 7 electric motor; 8 single stage gearbox; 9 frequency converter; 10 chain transmission; 11 support racks; 12 shaker racks; 13 connecting bracket; 14 support plate; 15 network filter.

When determining the qualitative indicators of separation, the following indicators were determined:

- damage to root crops and onions;
- completeness of separation of a heap of root crops and onions.

Laboratory studies were carried out depending on the various options for the location of the angle of inclination of the rod elevator and conducted in the following sequence [8].

The onion-soil pile was prepared in accordance with the fractional composition of the onion-sowing roll during its selection (onion-sowing of the "Stuttgart Riesen" variety, with soil moisture of 18%).

The composition of the heap and the main physical and mechanical properties of its components were selected based on the materials of field research conducted with the participation of the author of the work during 2016–2017 in JSC "Ozery" of the Moscow region:

- root vegetables (bulbs), 65%;
- impurities, 35%—including:
- vegetable impurities, 5%;
- small soil impurities-15%;
- lumps of soil commensurate in size with the standard fraction of root crops (bulbs) in transverse size, 10%;
- lumps of soil having a larger transverse size, 5%.

A shaker (4) (elliptical and three-shouldered) was installed under the canvas of the rod elevator 3.

Next, a prepared pile of root crops (bulbs) was delivered to the canvas of the rod elevator (3) from the container (2) for the preliminary placement of the pile.

Laboratory studies were carried out in order to determine the optimal technological parameters of a rod elevator with an adjustable angle of inclination of the web in laboratory conditions.

During the study of the separation process of the onion-sowing pile, factors were identified, the total number of which was initially 15, which covered the technological and structural parameters of the rod elevator, as well as the physico–mechanical properties of the soil and onion-sowing bulbs.

Because it is impossible to cover the influence of all factors and their interactions during research, and on the basis of a priori information and specific research tasks, the most significant factors affecting the quality of onion-sowing separation were identified.

After initially selecting the most significant factors affecting the quality of onion separation, a screening experiment was conducted, according to the results of which, after processing, information about the significance of each parameter was obtained, which made it possible to exclude insignificant factors from further consideration and reduce the volume of further research.

To carry out the screening experiment, a matrix was made, taking into account the initially identified factors by randomly mixing two semi-replicas of type $2^{4-1}$ [10–12].

After the screening experiment, insignificant factors were eliminated and the three most significant factors affecting the quality of planting onion bulbs were established, namely:

- translational speed of movement of the web of the rod elevator $v_{ЭЛ}$, m/s;
- feeding of a pile of onions $Q_S$, kg/s;
- the angle of inclination of the rod elevator $\alpha_1$, deg.

For the optimization criterion, a qualitative indicator of harvesting was chosen—the completeness of the separation of bulbs $v$, %.

After processing the results based on the theory of a multifactorial experiment, using the computer program Statistic 10.0, the values of the response function were obtained—the completeness of the separation of bulbs with varying factors, in accordance with the second-order Box–Benkin plan and an adequate mathematical model, was obtained, describing the dependence of the completeness of separation $v = f(v_{EL}, Q_{Bp}, \alpha_1)$ of bulbs after their planting in the furrow in an encoded form on the selected factors:

$$Y = 98.86 + 0.15x_1 + 0.27x_2 + 0.12x_3 - 0.75x_1^2 - 0.35x_2^2 - 0.3x_3^2 + 0.1x_1x_2 + 0.09x_1x_3 + 0.25x_2x_3 \qquad (7)$$

where Y is the optimization criterion, %; $x_1$ is the translational speed of the web of the rod elevator, m/s $x_2$ is the supply of a pile of onions, and kg/s $x_3$ is the angle of inclination of the rod elevator, deg.

The results of the multifactorial experiment were processed using the computer program "STATISTICA-10.0"; as a result, the values of the response functions were obtained—the completeness of the separation of a pile of onions with varying factors, in accordance with the second-order Box–Benkin plan and an adequate mathematical model, was obtained, describing the dependence of the completeness of the separation of a pile of onions $\nu = f(v_{EL}, Q_S, \alpha_1)$ in an encoded form on the selected factors:

The hypothesis of the adequacy of the second-order model was tested by statistical analysis of the regression equation.

The results of the calculation of statistical characteristics are presented in Figure 12.

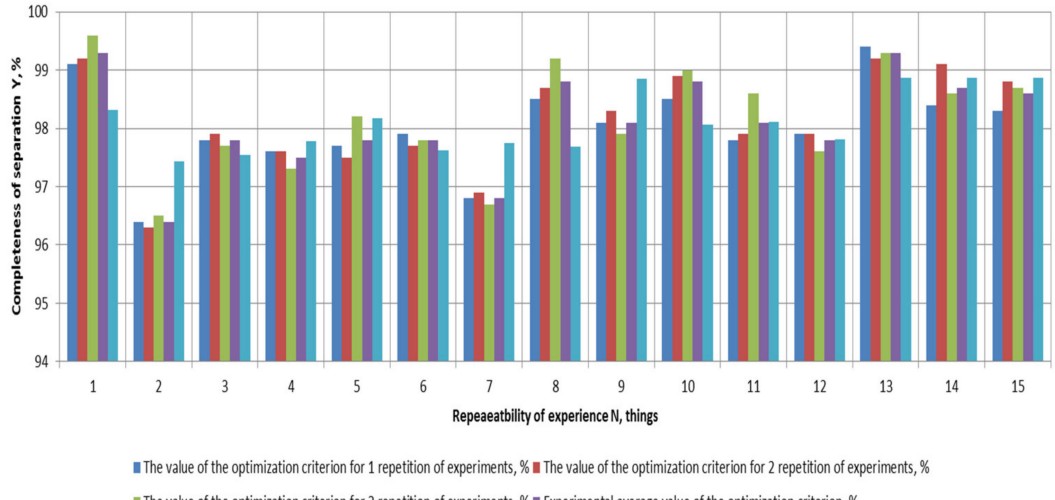

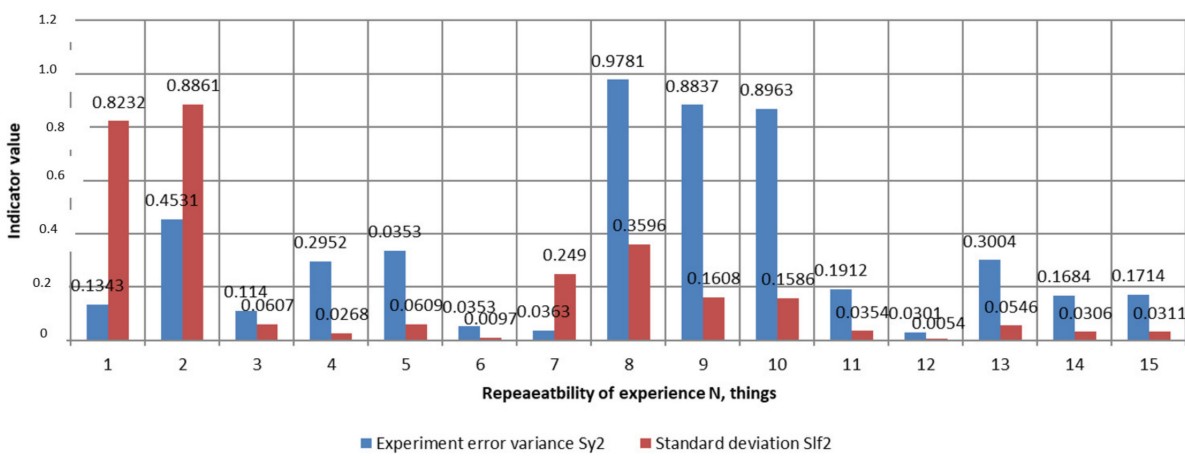

**Figure 12.** Statistical characteristics of experience error.

The value of the Fisher criterion Ft at the 5% significance level for the resulting equation with degrees of freedom of the numerator $\nu_1 = N_o - (k_\varphi + 1) = 11$ and the denominator $\nu_2 = N_o (m - 1) = 30$, selected according to the table, is 2.

The calculated value of the Fisher criterion F = 1.97. Since $F_T = 2.1 > F = 1.97$, we get an adequate mathematical model.

Equation (1) in decoded form has the following form:

$$Y = 41.41 + 68.01v_{EL} - 0.01Q_S + 0.26\alpha_1 - 21.64v_{EL}^2 - 0.04Q_S^2 - 0.09\alpha_1^2 + 0.12v_{EL}Q_S + 12.21v_{EL}\alpha_1 - 0.54Q_S\alpha_1 \quad (8)$$

Substituting the value $x_1 = 0$ into Equation (8), we obtain a two-dimensional cross-section of the response surface characterizing the quality indicator of the separation of the pile of onions from the supply of the pile ($x_2$) and the angle of inclination of the rod elevator, deg ($x_3$):

$$Y = 98.86 + 0.27x_2 + 0.12x_3 - 0.35x_2^2 - 0.3x_3^2 + 0.25x_2x_3. \quad (9)$$

The coordinates of the center of the response surface are determined by differentiating Equation (3) and solving a system of equations:

$$\begin{cases} \frac{dy}{dx_2} = 0.27 - 0.7x_2 + 0.25x_3 = 0, \\ \frac{dy}{dx_3} = 0.12 - 0.6x_3 + 0.25x_2 = 0. \end{cases} \quad (10)$$

Solving the system of Equation (10), we find the coordinates of the center of the surface of the response function in encoded form: $x_2 = 0.537$; $x_3 = 0.423$ (in decoded form $Q_S = 28.2$ kg/s, $\alpha_1 = 18.3$ degrees).

Substituting the values $x_2$ and $x_3$ into Equation (10), we obtain the value of the response function in the center of the surface:

$$Y_S = 98.74 \quad (11)$$

After performing the canonical transformation of Equation (10), we obtain the equation in canonical form:

$$Y_S - 98.74 = -0.05x_2^2 - 0.51x_3^2 \quad (12)$$

The angle of rotation of the axes will be:

$$tg2\alpha_2 = \frac{-0.25}{-0.35 - 0,31} = 5.3 \quad (13)$$

Angle $\alpha_2 = 68°$.

Substituting different values of the response function into Equation (2), the equations of contour curves–ellipses were obtained. The calculation results are shown in Figure 13.

Figure 11 shows that the completeness of the separation of the pile of onions on a rod elevator with an adjustable angle of inclination of the web is 98% when finding the optimal values of the factors under consideration: the supply of the pile of onions $Q_S = 18.6$ 38.4 kg/s and the angle of inclination of the web a_1 = 13.3 . . . 23.2 degrees.

The two-dimensional cross-section of the response surface characterizing the quality of separation of the onion pile from the translational speed of the rod elevator ($x_1$) and the angle of inclination of the web ($x_3$) is described by Equation (1) at $x_2 = 0$, after which:

$$Y = 98.86 + 0.15x_1 + 0.12x_3 - 0.75x_1^2 - 0.3x_3^2 + 0.09x_1x_3 \quad (14)$$

When differentiating Equation (14), we obtained a system of equations:

$$\begin{cases} \frac{dy}{dx_1} = 0.15 - 0.14x_1 + 0.09x_3 = 0, \\ \frac{dy}{dx_3} = 0.12 - 0.6x_3 + 0.09x_1 = 0. \end{cases} \quad (15)$$

Based on the system of equations, the coordinates of the center of the response surface were obtained: $x_1 = 1.05$; $x_3 = -0.115$ (respectively, in decoded form $v_{EL} = 1.64$ m/s, $\alpha_1 = 16.7$ degrees).

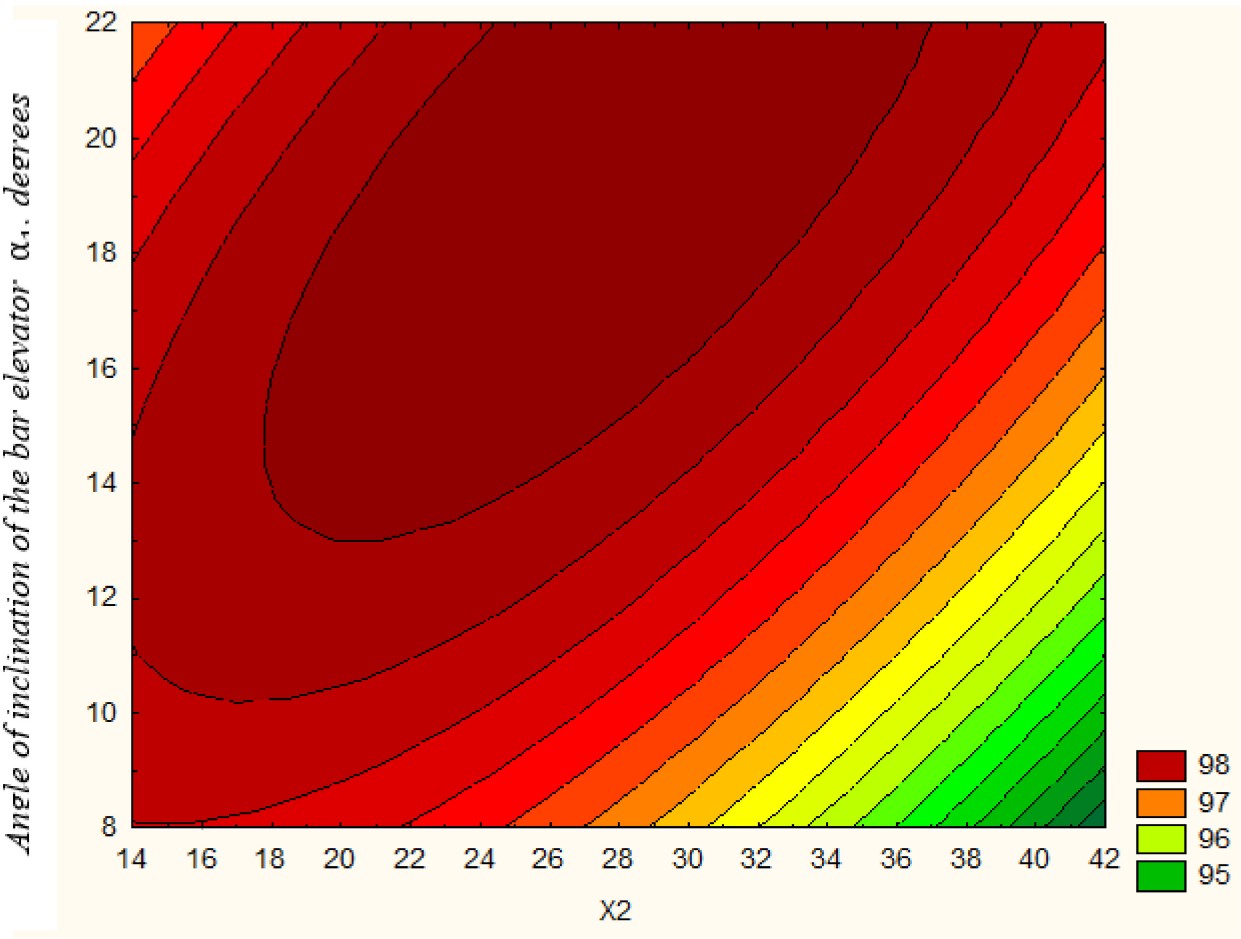

**Figure 13.** Two-dimensional cross-section of the response surface, characterizing the dependence of the completeness of the separation of the pile of onions on the supply of the pile of onions (kg/s) and the angle of inclination of the web of the rod elevator (deg).

Substituting the values $x_1$ and $x_3$ into Equation (15), we obtain the value of the response function in the center of the surface:

$$Y_S = 98.28 \tag{16}$$

After calculating the coefficients of the regression equation in canonical form, the regression equation will be written in the canonical form:

$$Ys - 98.28 = 0.71x_1^2 - 0.92x_3^2 \tag{17}$$

The angle of rotation of the axes will be:

$$tg2\alpha_2 = \frac{0.09}{0.7 - 0.09} = 0.147 \tag{18}$$

Angle $\alpha_2 = 4.3°$.

A two-dimensional cross-section of the response surface was built based on the obtained data (Figure 13).

Analyzing Figure 14 shows that the completeness of the separation heap of onions rod the Elevator with adjustable angle of incline is 98% at finding the optimal values of the considered factors—the translational speed of movement of a cloth rod Elevator and the angle of incline rod Elevator is within $\alpha_1 = 13.9\ 18.4\ldots$ hail, $v_{EL} = 1.54\ 1.69$ m/s.

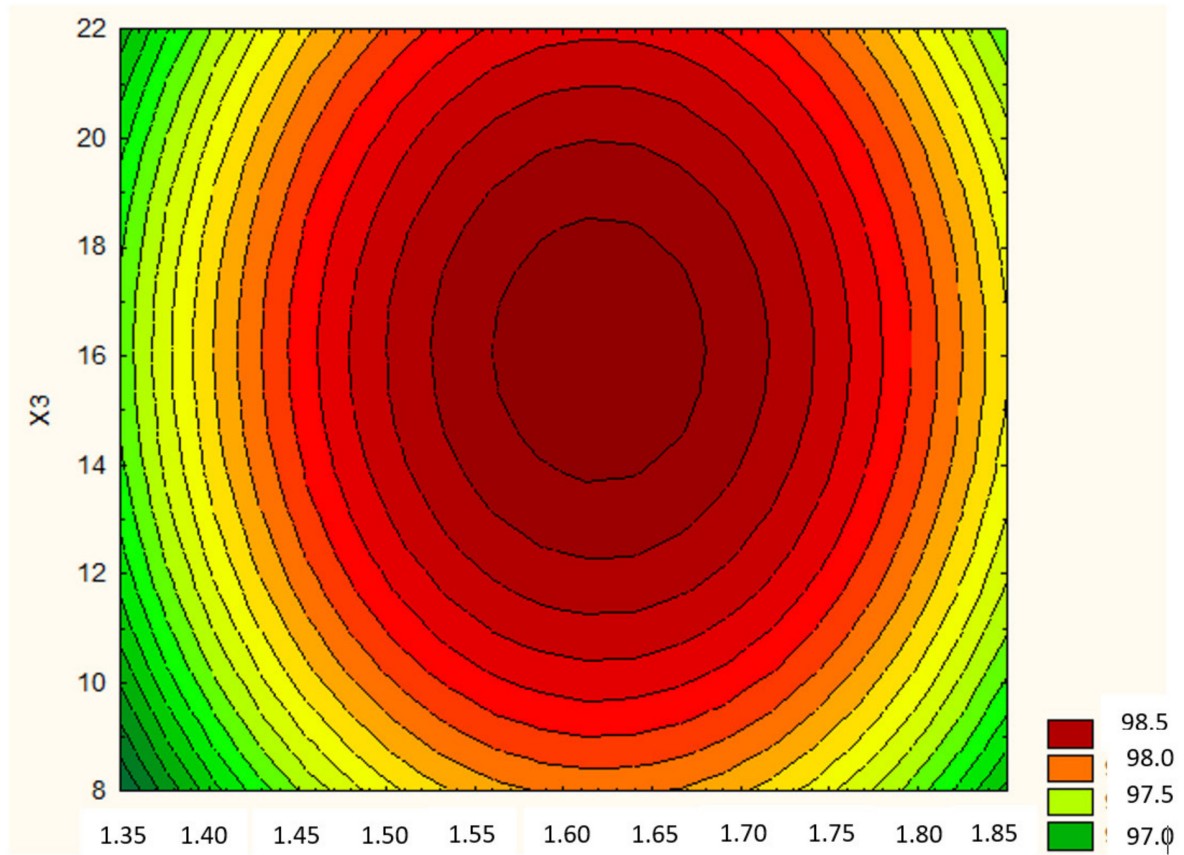

*Pile of onions on the translational speed of the web of the rod elevator $V_{EL}$, m/s*

**Figure 14.** Two-dimensional cross-section of the response surface, characterizing the dependence of the completeness of the separation of the pile of onions on the translational speed of the web of the rod elevator (m/s) and the angle of inclination of the web of the rod elevator (deg).

The two-dimensional cross-section of the response surface characterizing the quality of separation of the onion pile from the translational speed of the rod elevator ($x_1$) and the supply of the onion pile ($x_2$) is described by Equation (15) at $x_3 = 0$ after which:

$$Y = 98.86 + 0.15x_1 + 0.27x_2 - 0.75x_1^2 - 0.35x_2^2 + 0.1x_1x_2 \qquad (19)$$

Differentiating Equation (13) and solving the system of equations:

$$\begin{cases} \frac{dy}{dx_1} = 0.15 - 1.5x_1 + 0.1x_2 = 0, \\ \frac{dy}{dx_2} = 0.27 - 0.7x_2 + 0.1x_1 = 0, \end{cases} \qquad (20)$$

We get the coordinates of the response surface $x_1 = 0.12$, $x_2 = 0.4$ ($v_{EL} = 1.63$ m/s, $Q_S = 23.8$ kg/s).

Substituting the values $x_1$ and $x_2$ into Equation (20), we obtain the value of the response function in the center of the surface:

$$Y_S = 98.98 \qquad (21)$$

After calculating the coefficients of the regression equation in the canonical form, it will be written:

$$Y_S - 98.98 = 0.125x_1^2 - 0.575x_2^2 \qquad (22)$$

The angle of rotation of the axes will be:

$$\text{tg}2\alpha_2 = \frac{0.1}{0.75 + 0.35} = 0.09 \tag{23}$$

Angle $\alpha_2 = 2.3°$.

Analyzing Figure 15, the completeness of the separation of the onion pile is 98% when finding the optimal values of the factors under consideration: the translational speed of the web of the rod elevator and the supply of the onion pile $v_{EL} = 1.55 \ldots 1.68$ m/s, $Q_s = 17.9 \ldots 28.3$ kg/s.

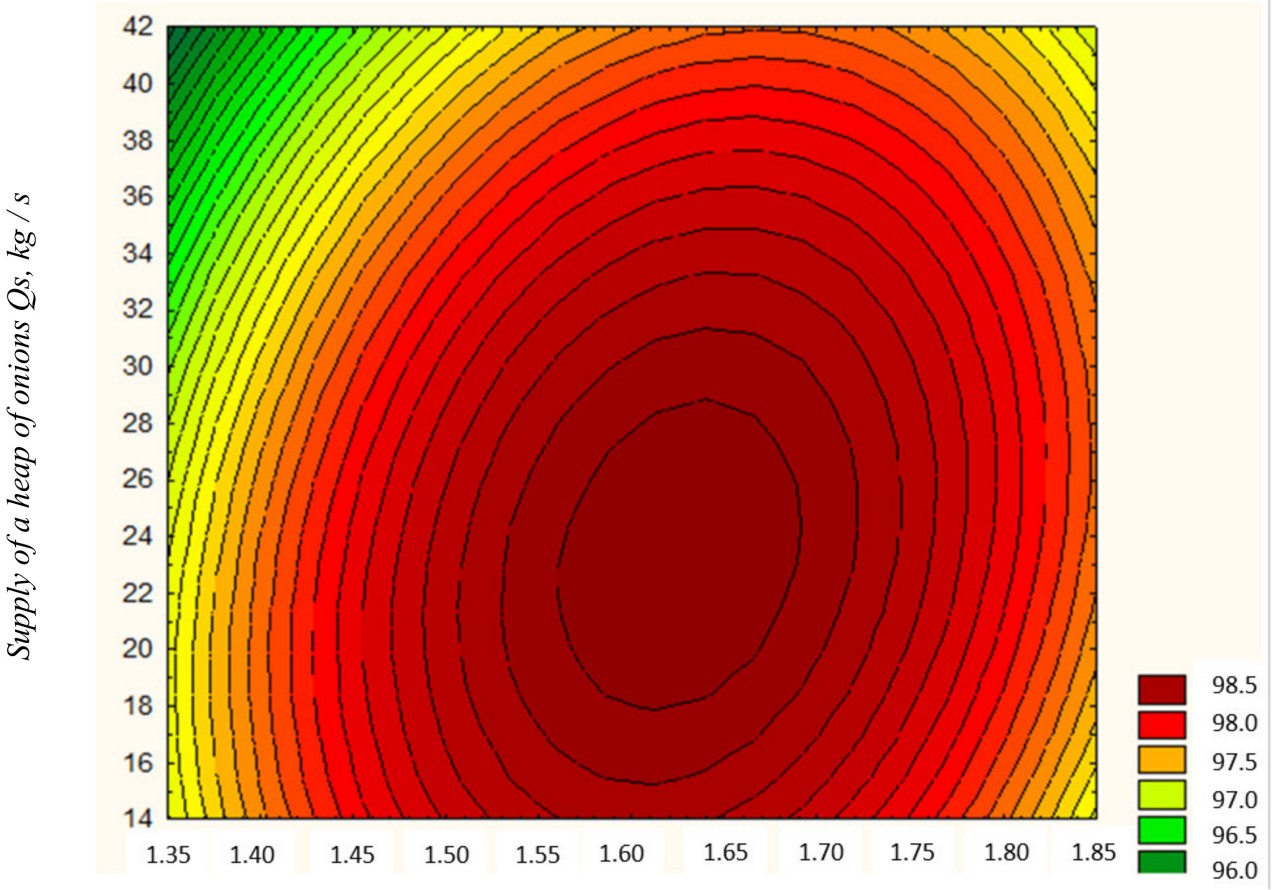

*Pile of onions on the translational speed of the web of the rod elevator $V_{EL}$, m/s*

**Figure 15.** Two-dimensional cross-section of the response surface, characterizing the dependence of the completeness of the separation of the onion pile on the translational speed of the web of the rod elevator (m/s) and the supply of the onion pile (kg/s).

Equation (8), taking into account the significance of the regression coefficients, can be presented in the following form:

$$Y = 41.41 + 68.01v_{EL} - 0.01Q_S + 0.26\alpha_1 - 21.64v_{EL}^2 - 0.04Q_S^2 - 0.09\alpha_1^2 + 0.12v_{EL}Q_S + 12.21v_{EL}\alpha_1 - 0.54Q_S\alpha_1 \tag{24}$$

The general view of the machine for the developed technology of harvesting root crops and onions is shown in Figure 16.

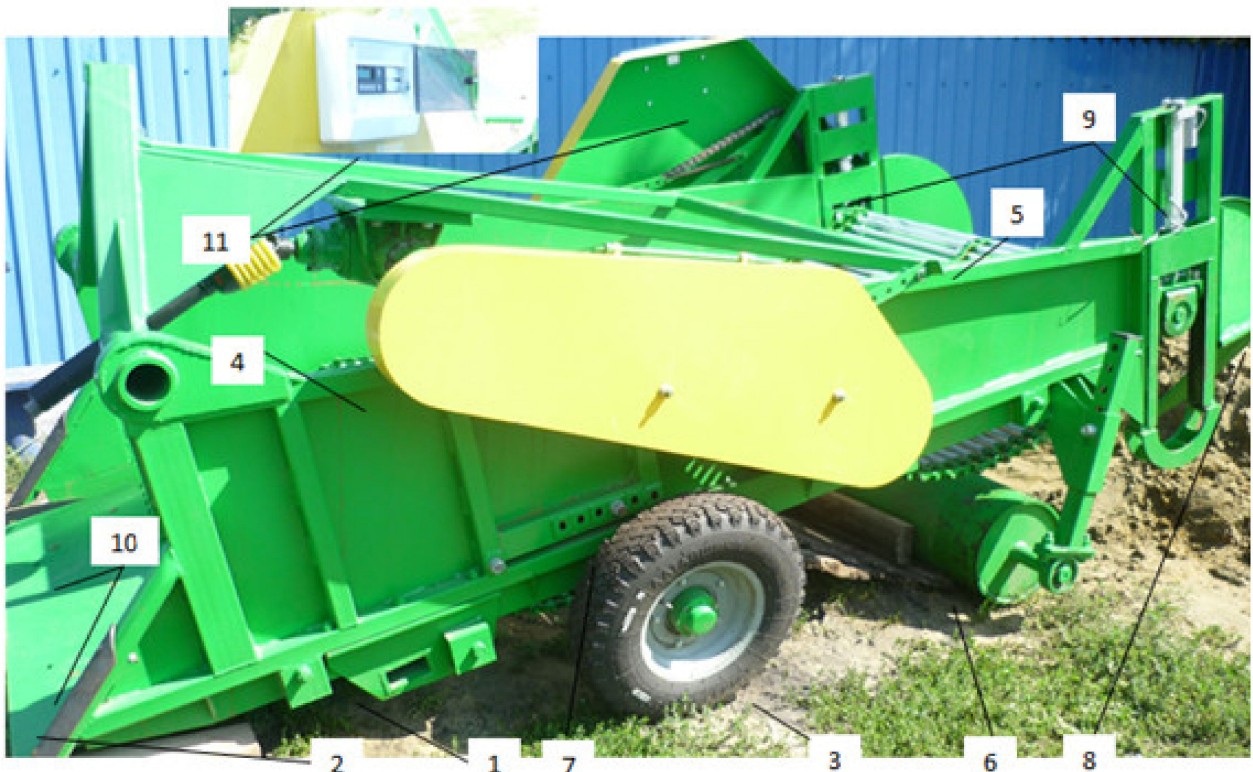

**Figure 16.** General view of a machine for harvesting root crops and onions, equipped with a separating rod elevator with an adjustable angle of inclination of the web: 1, frame; 2, digging ploughshare; 3, support wheels; 4, main separating rod elevator; 5, additional rod elevator; 6, roller-bed-forming machine; 7, adjusting plate of the shaker; 8, narrowing tray; 9, electric cylinders; 10, weight sensor of the digging ploughshare; 11, microcontroller.

## 4. Conclusions

Therefore, based on theoretical and experimental studies, an algorithm and software for the control system for the drive of the working bodies of the machine for harvesting root crops and onions have been developed; in addition, a machine for harvesting root crops and onions has been developed, equipped with a separating rod elevator with an adjustable angle of inclination of the web, which makes it possible to improve the quality indicators of harvesting root crops and onions.

A constructive and technological scheme of a separating bar elevator with an adjustable angle of inclination of the web, providing an increase in the quality indicators of onion harvesting, has been substantiated and developed. The analysis of the system of Equation (11), reflecting the relative velocity of the onion-sowing pile element along the surface of the bar elevator in the longitudinal direction, indicates that with an increase in the angle a_1 of the longitudinal slope of the web, the relative velocity $v_{Mr}$ of the onion-sowing pile movement decreases. In the absence of airflow, the relative velocity $v_{Mr}$ of the movement of the pile of onion-sowing when the angle of inclination of the web of the bar elevator changes in the longitudinal direction in the considered range varies from 0.34 to 0.47 m/s. Changing the angle $\alpha_1$ of the longitudinal inclination of the rod elevator by 5° significantly $v_{Mr}$ changes the relative average speed of the onion-sowing pile, which improves its quality of separation without increasing the translational speed of the rod web, thus leading to increased bulb damage as a result of the intense impact of the rod elevator web shakers on the processed onion pile.

The developed mathematical model of the movement of a pile of onion-sowings on the surface of a bar elevator when changing the direction of its oscillations in the horizontal plane is represented by a system of Equation (19).

**Author Contributions:** Conceptualization, M.N.E. and A.S.D.; methodology, A.S.D. and M.M.G.; software, N.V.S. and M.A.M.; validation, M.A.M.; investigation, A.G.A.; resources, M.M.G.; writing—original draft preparation, N.V.S.; writing—review and editing, A.V.S.; project administration, A.G.A.; funding acquisition, A.S.D. All authors have read and agreed to the published version of the manuscript.

**Funding:** This work was supported by a grant from the Ministry of Science and Higher Education of the Russian Federation for large scientific projects in priority areas of scientific and technological development (grant number 075-15-2020-774).

**Institutional Review Board Statement:** Not applicable.

**Informed Consent Statement:** Not applicable.

**Data Availability Statement:** The raw data supporting the conclusions of this article will be made available by the authors, without undue reservation.

**Conflicts of Interest:** The authors declare that they have no known competing financial interests or personal relationships that could have appeared to influence the work reported in this paper.

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
