# Peer review of "Development and Modeling of an Onion Harvester with an Automated Separation System"

_agriengineering, doi:10.3390/agriengineering4020026_

Round 1

Reviewer 1 Report

The paper presents a development and modeling method of an onion harvester with automated separation system and submitted to AgriEngineering, however, the header and footnote is Sustainability Journal.
The following are some humble comments to the paper:

1. Abstract:
Please rewrite the Abstract to be more concise. Current Abstract has 442 words and it can be reduced to 250 - 300 words.
2. Introduction:
The Introduction can be added with more content such as existing problem, aims of the study, etc. Current Introduction only has 2 paragraphs.
3. Literature review:
- Figures 1-3 were not sharp enough, please replace with a better resolution.
- Figure 4 also blurring and a bit difficult to see, please replace with a sharper image.
- What is the symbol in line 129? I did not see any Equation related to this math symbol.
- Is Figure 5 obtained from another publication? Please provide a citation number as well as a copyright permission to use the Figure.
- Line 276, missing an Equation number. It supposed to be (5)
4. Results and Discussion:
- I don't thing Figure 9 is appropriate to be placed in Results and Discussion section.
- Lines 332 onward, I noticed that this Section include the sequence of the laboratory experiment. I would suggest Authors to create a flowchart for representing these sequence so that it can be understand clearly.
- I suggest to have a Figure or Chart from the statistic result because Figure or Chart is better presentation and it will be helpful to complement the result presented in Table.
- Figures 11, 12, and 13, missing x-axis caption. Please revise the Figure.

General comments:
1. Please double check the font size of all Equations as it is not uniform.
2. Please present the Figures proportionally to the text and provide sharp images.
3. Please provide a Authors Contribution in the paper.

Author Response

Reviewer â„– 1, Remark â„– 1:

  1. The sections "Introduction" and "Literature Review" have been corrected, the goals and objectives of the research are presented.

Reviewer â„– 1, Remark â„– 2:

  1. Sections "Introduction" and "Literature Review" are combined into one section. Relevant studies are presented with the corresponding trends in the development of machine technologies.

Reviewer â„– 1, Remark â„– 3:

  1. Research results are presented in the literature review section

Reviewer â„– 1, Remark â„– 4:

  1. The sections "Introduction" and "Literature Review" are combined into a single section and the purpose and novelty of the research are presented.

Reviewer â„– 1, Remark â„– 5:

  1. The section "Materials and Methods" presents the area of research.

Reviewer â„– 1, Remark â„– 6:

  1. The methodology provides information on target system optimization

Reviewer â„– 1, Remark â„– 7:

  1. The list of literature presents patent No. 2638190 and patent No. 2679734.

Reviewer â„– 1, Remark â„– 8:

  1. The text contains links to studies on changes in cleaning quality indicators when varying the angle of inclination of the rod elevator web.

Reviewer â„– 1, Remark â„– 9:

  1. The text with line numbers 207-210 are patent citations.

Reviewer â„– 1, Remark â„– 10:

  1. Substantiation of the model developed in the study.

Reviewer â„– 1, Remark â„– 11:

  1. The substantiation of the model developed during the research is carried out.

Reviewer â„– 1, Remark â„– 12:

  1. Images of models are presented in the form of two-dimensional sections, reflecting the 3D model.

Reviewer â„– 1, Remark â„– 13:

  1. The engineering equations presented in the article are displayed in accordance with the format.

Reviewer â„– 1, Remark â„– 14:

  1. Typographical errors have been improved.

Reviewer â„– 1, Remark â„– 15:

Limitations of the study, as well as suggestions for further research, are included in the final section of the revised version of the article.

Reviewer 2 Report

The authors presented a study as “Development and modelling of onion harvesters with automated separation system”. The idea of the research is good, and some suggestion is given to improve the manuscript before publication publishable in the respective journal.

The abstract of the first three to four sentences are so length so need to be modify and irrelevant sentences can be remove.  The authors are requested to revise the abstract with clear findings and results.

  1. The introduction section is too short contains unnecessary information, it should be concise focusing on background, targeted problem, available solutions, research gap and objectives of the study. Refences of the quantitative values also missing i.e., values use in the L39&40.
  2. Introduction and review of literature section can be in one unit so a comprehensive problem statement can emerge. The authors are requested to include some relevant and latest research with respective trends which directly relate to the research situation cited in the paper so that the reader gets an overview of the end conclusions.
  3. There is no need to use figures in the literature review section, while is should be more proficient if the authors use methodology, finding and conclusion of the previous studies about the onion harvesters i.e., Asghar et.al [1] design a groundnut digger, finding of that study was that, and conclusion was that etc.
  4. It should be more suitable if the authors merge the introduction, literature review and novelty and aim of the study in single section, it should be easily understandable for the readers.
  5. In material and method section study area is missing.
  6. Methodology is very poor and do not provide enough information about the optimization of the purposed system.
  7. In the material and methods section L171&172 the authors code the two patents (patent 171 No. 2638190) & (patent No. 2679734) without reference.
  8. The authors selected rod elevator angle between 15-230 without any reference, while in the literature review, they are not discussing rod elevation angle effect studies on the harvesting of crops.
  9. Paragraph containing L207-L210 needs reference.
  10. The authors do not include the justification part of the specific model developed in the research study.
  11. The justification part of the specific model developed in the research study is very difficult to understand, it should be explained below the labeled figure.
  12. Design models pictures should be in the 3D view which should helpful for the manuscript strengthening.
  13. Through all manuscript engineering Equations used without any reference and not according the format.
  14. The typographical errors and language could be improved to convey the complete idea of the research paper concisely.
  15. Limitations of the study, as well as suggestions for further research, are included in the concluding section of the revised version of the paper. This will strengthen the quality of the paper.

Author Response

Reviewer â„– 2, Remark â„– 1: Section "Abstract" rewritten.

Reviewer â„– 2, Remark â„– 2: Introduction supplemented with additional content

Reviewer â„– 2, Remark â„– 3: Can't get better image resolution.

Reviewer â„– 2, Remark â„– 4: A flowchart has been created to represent the sequence, the execution of the research.

Reviewer â„– 2, Remark â„– 5: A diagram of the statistical characteristics of the experimental error is presented.

Reviewer â„– 2, Remark â„– 6: Figures 11, 12 and 13 are labeled along the X axis.

Reviewer â„– 2, Remark â„– 7: The font size of all equations has been adjusted.

The figures in the article are made in proportion to the text.

The contribution of the authors to the article is indicated.

Reviewer 3 Report

The authors of the work took up the topic related to reducing crop loss during harvesting.

However, from the abstract of the work one could conclude something completely different, namely, that the work was devoted to economic and not technical aspects. 

In my opinion, work in the current version cannot be allowed to print. Authors need to correct the introduction and describe more specifically what they do in the work. In addition, the summary also does not fully reflect the content of the article.

Another note is the drawings. Figure 3 is not readable to me, in addition, I think that Figure 5 has an inadequate size to the data contained - it should be smaller. Figure 6 should be signed on the page where it is placed and not on the next one and I think that by reducing its size we will not lose quality. Especially comparing it with Figure 9 which is aesthetic and legible. Figures 11, 12 and 13 do not have horizontal axis captions and Figure 13 also has vertical axes.

Another issue of their patterns and symbols their font makes them look very unsightly. Poems: 129, 140, 143, 145, 146, 150, 151, 235, 240, 241, 242, 243, 245, 247, 248, 249, 270, 271, 272, 276, 283, 284, 320, 374, 375, 376, 378, 383, 385, 386, 395, 401, 402, 407, 408, 413, 416, 417, 419, 420, 423, 426, 428, 444, 446, 448, 452, 455, 457, 464, 473, 475, 476, 477, 480,  483, 485, 486, 497, 498, 519, 521, 523, 524.
All symbols in Table 1. Symbols in the header of Table 2.

In line 320, the second value of 220 V is written correctly, while the "=" sign appears twice next to it.

In the list of literature, the authors give 30 items, but in the text there is a reference to only 13.

In line 439, the incorrectly written units should be kg/s

Author Response

Reviewer â„– 3, Remark â„– 1: The presented research results were corrected.

Round 2

Reviewer 1 Report

Dear Authors,

Thank you for providing the revised version and considering the Reviewer's comments. I have checked the revised documents and I have no further comments.

Kind regards,

- Reviewer 1 -

Author Response

Hello dear reviewers! I am sending you a revised version of the article.

Reviewer 3 Report

After replying to my comments. The work may be published.

Author Response

(The authors gave the same response as above.)
